# OsMYB58 Negatively Regulates Plant Growth and Development by Regulating Phosphate Homeostasis

**DOI:** 10.3390/ijms25042209

**Published:** 2024-02-12

**Authors:** Dongwon Baek, Soyeon Hong, Hye Jeong Kim, Sunok Moon, Ki Hong Jung, Won Tae Yang, Doh Hoon Kim

**Affiliations:** 1Plant Molecular Biology and Biotechnology Research Center, Gyeongsang National University, Jinju 52828, Republic of Korea; dw100@hanmail.net; 2National Agrobiodiversity Center, National Institute of Agricultural Sciences, Rural Development Administration, Jeonju 54874, Republic of Korea; hsy1203@korea.kr; 3College of Life Science and Natural Resources, Dong-A University, Busan 49315, Republic of Korea; hjkim83@dau.ac.kr; 4Graduate School of Biotechnology and Crop Biotech Institute, Kyung Hee University, Yongin 17104, Republic of Korea; moonsun@khu.ac.kr (S.M.); khjung2010@khu.ac.kr (K.H.J.)

**Keywords:** MYB transcription factor, phosphate starvation signaling, *OsMYB58*, *OsmiR399*, PHR-miR399-PHO2

## Abstract

Phosphate (Pi) starvation is a critical factor limiting crop growth, development, and productivity. Rice (*Oryza sativa*) R2R3-MYB transcription factors function in the transcriptional regulation of plant responses to various abiotic stresses and micronutrient deprivation, but little is known about their roles in Pi starvation signaling and Pi homeostasis. Here, we identified the R2R3-MYB transcription factor gene *OsMYB58*, which shares high sequence similarity with *AtMYB58*. *OsMYB58* expression was induced more strongly by Pi starvation than by other micronutrient deficiencies. Overexpressing *OsMYB58* in *Arabidopsis thaliana* and rice inhibited plant growth and development under Pi-deficient conditions. In addition, the overexpression of *OsMYB58* in plants exposed to Pi deficiency strongly affected root development, including seminal root, lateral root, and root hair formation. Overexpressing *OsMYB58* strongly decreased the expression of the rice microRNAs *OsmiR399a* and *OsmiR399j*. By contrast, overexpressing *OsMYB58* strongly increased the expression of rice *PHOSPHATE 2* (*OsPHO2*), whose expression is repressed by miR399 during Pi starvation signaling. OsMYB58 functions as a transcriptional repressor of the expression of its target genes, as determined by a transcriptional activity assay. These results demonstrate that OsMYB58 negatively regulates *OsmiR399*-dependent Pi starvation signaling by enhancing *OsmiR399s* expression.

## 1. Introduction

Phosphorus (P) in the soil is an essential macronutrient for crop growth, development, and reproduction [1]. P is associated with plant respiration and photosynthesis because they are key components in several basic composition biomolecules, such as nucleotide, ATP, NADPH, and phospholipids [2]. The P amounts for plant growth and development need much larger amounts than the available P in soil [3]. The inorganic P (Pi; orthophosphate) is the unique form that can absorbed by plant roots [1]. Pi deficiency is one of the important nutrient deficiencies caused by limiting plant productivity [4]. To adapt against Pi deficiency, plants have evolved numerous physiological or molecular regulatory mechanisms, such as remodeling of the root structures, changes in metabolic processes, and the regulation of gene expression [5].

MYB transcription factors, which are widespread in crops, play important roles in regulating plant signaling pathways in response to various stresses [6,7]. MYB transcription factor classes are divided into the 1R-MYB, 2R-MYB (R2R3-MYB), 3R-MYB, and 4R-MYB classes [8]. Functional studies have revealed that R2R3-MYB transcription factors function in a wide variety of plant responses, including stress tolerance, plant hormonal regulation, metabolism, development, and cell differentiation [1,9]. R2R3-MYB transcription factors in plants function in transcriptional regulation in response to diverse abiotic stresses, such as salt stress, drought, cold stress, abscisic acid (ABA) treatment, oxidative damage, and phosphate (inorganic phosphate; Pi) deficiency [9,10,11,12,13,14,15]. Overexpressing *OsMYB2* in rice (*Oryza sativa* L.) enhanced plant tolerance to salt, cold, and drought stress [10]. TaMYB31-B in hexaploid wheat (*Triticum aestivum* L.) positively regulates plant responses to drought by regulating the expression of wax biosynthesis genes and drought-responsive genes [11]. TcMYB29a in *Taxus chinensis* is a transcriptional activator that functions in ABA-mediated signaling by binding to MYB-recognizing element (MRE) motifs in the promoters of taxol-biosynthesis-related genes [12]. AtMYB30 in *Arabidopsis thaliana* is involved in systemic reactive oxygen species (ROS) signaling in response to high-light stress [13]. The transcription factor TaMYB4-7D in *T. aestivum* mediates Pi uptake and translocation under Pi-deficient conditions by binding to four MYB-binding site (MBS) motifs in the promoter region of *TaPHT1;9-4B* [14]. The interaction of PuMYB40 and PuWRKY75 in *Populus ussuriensis* positively regulates adventitious root formation under low-Pi conditions [15].

Rice contains 99 R2R3-MYB transcription factor genes, accounting for 52.1% of the 190 MYB genes [16]. A few R2R3-MYB transcription factors in rice were recently shown to be associated with Pi-starvation responses. For instance, OsMYB2P-1 functions as a key Pi-dependent regulator of Pi-starvation signaling by controlling the expression of Pi transporter (PT) genes, such as *OsPT2*, *OsPT6*, *OsPT8*, and *OsPT10* [17]. OsMYB4P acts as a transcriptional activator of Pi-homeostasis-related genes to increase Pi acquisition in rice [18]. Overexpressing *OsMYB4P* strongly induced the expression of PT genes, including *OsPT1*, *OsPT2*, *OsPT4*, *OsPT7*, and *OsPT8*, in shoots under Pi-deficient conditions [18]. OsMYB1 mediates root elongation in the Pi starvation responses and acts as an important regulator of both Pi-starvation signaling and gibberellic acid (GA) biosynthesis [19]. Overexpressing *OsMYB5P* enhanced plant tolerance in terms of Pi deficiency by controlling the transcription of PT genes [20].

Pi is an important macronutrient for plant growth and development and a component of various biological molecules such as nucleic acids, membrane lipids, and ATP [21,22]. These molecules function as essential regulators of Pi-starvation signaling and cellular mechanisms to systemically enhance Pi uptake, transport, and utilization [23]. Pi-starvation signaling is mediated by the PHR-miR399-PHO2 molecular network [21,24,25]. The rice *miRNA399* (*OsmiR399*) family comprises 11 members (*OsmiR399a* to *k*) that are important regulators of Pi-starvation signaling [25]. During exposure to Pi-deficient stress, the expression of *OsmiR399s* is widely induced, whereas the expression of their target gene *OsPHO2* ultimately decreases [21,25]. *OsPHR1* and *OsPHR2*, the homologs of Arabidopsis *PHR1*, encode transcription factors that positively regulate the expression of *OsIPS1* [24,26]. *OsIPS1* represses the activity of *OsmiR399s* via a target mimicry mechanism [24,26]. The OsPHO2 perceives the OsmiR399s moving from the shoot to the root and contributes to the throughput of Pi from the root to the shoot in Pi-starvation signaling [25]. In Pi-starvation signaling, the PHR-miR399-PHO2 systemic regulatory network influences the activity of two important Pi transporters, PHO1 and PT2, to help maintain Pi homeostasis [27,28]. In addition, the OsPHR2-OsmiR827-OsSPX regulatory network is associated with Pi-starvation signaling in rice [24,26]. *OsmiR827* is highly expressed under Pi deficiency, which induces *OsmiR827* expression, thereby decreasing the expression of *OsSPX* genes [26].

Pi deficiency leads to changes in plant morphology, physiology, and biochemistry by decreasing the acquisition and utilization efficiency of Pi from the soil [1,5]. Although the functions of many R2R3-MYB transcription factors in plant responses to abiotic stress have been extensively investigated, more research is needed on their roles in Pi-deficient responses. In this study, we characterized the functions of rice *OsMYB58* in PHR-miR399-PHO2-dependent Pi starvation signaling. Through physiological and biological analyses, we demonstrated that OsMYB58, which functions as a transcriptional repressor essential for *OsmiR399s* expression, plays a key role in maintaining Pi homeostasis. Current agriculture is heavily dependent on P fertilizer to provide against the low Pi availability in soils [29]. Because of the properties of P fertilizer and the low efficiency of plant P absorbability, there is crucial research and development focused on producing crops that have enhanced P use efficiency [29]. Understanding the role of *OsMYB58* in PHR-miR399-PHO2-dependent mechanisms during Pi-starvation signaling will help enhance crop yields and growth in nutrient-poor soils.

## 2. Results

### 2.1. OsMYB58 Expression Is Enhanced under Phosphate-Deficient Conditions

We previously isolated and characterized several R2R3-type MYB transcription factors that function in phosphate starvation signaling [18,20]. In the current study, to identify Pi-deficiency-related R2R3-type MYB transcription factors in rice, we investigated the expression levels of several R2R3-type OsMYB transcription factor genes in wild-type rice plants under various nutrient deficiency conditions. Among the many R2R3-type MYB genes with high expression under nutrient-deficient conditions, *OsMYB58* was highly expressed in rice under Pi-deficient conditions (Figure 1a, Appendix A). The full protein sequence of OsMYB58 (*LOC_Os02g46780*) shared high levels of sequence similarity with two Arabidopsis two genes, *AtMYB58* (*AT1G16490*; 60.5% similarity) and *AtMYB63* (*AT1G79180*; 59.1% similarity) (Figure 2a) [30]. The R2R3 MYB transcription factor OsMYB58 was classified in the same clade as OsMYB63 (*LOC_Os04g50770*) via phylogenetic analysis (Figure 2b). OsMYB58 is homologous to OsMYB63, with 72% sequence similarity, as revealed using the Phytozome database (https://phytozome-next.jgi.doe.gov/ (accessed on 11 January 2023)) [31]. To investigate the changes in *OsMYB58* expression in rice shoots and roots, we extracted total RNA from the shoots and roots of plants under high-Pi and low-Pi conditions. The expression levels of *OsMYB58* were similar in shoots and roots under high-Pi conditions, whereas it was expressed at higher levels in roots than in shoots under low-Pi conditions (Figure 1b). In the shoot, *OsMYB58* expression showed a clear increase, followed by a decrease, in the course of 7 days of exposure to low Pi (Figure 1c). On the other hand, *OsMYB58* expression consistently increased in roots under low-Pi conditions (Figure 1d). These results suggest that OsMYB58 is associated with Pi-starvation signaling.

### 2.2. Heterologously Overexpressing OsMYB58 in Arabidopsis Disrupts Pi Homeostasis in Response to Pi Deficiency

To explore the roles of OsMYB58 in the Pi starvation response, we heterologously overexpressed *OsMYB58* in Arabidopsis Col-0 plants (Figure 3a). Using RT-PCR analysis, we selected two lines of *OsMYB58*-overexpressing Arabidopsis plants (*OsMYB58*-AraOX) with different *OsMYB58* expression levels for further analysis (Figure 3b). To investigate the phenotypic alterations of wild-type Col-0 and *OsMYB58*-AraOX plants in response to Pi deficiency, we subjected 4-day-old seedlings to Pi deficiency for 7 days. The shoots and roots of *OsMYB58*-AraOX plants grew more slowly than those of Col-0 plants under both high-Pi and low-Pi conditions (Figure 3c). In addition, the shoots and roots of *OsMYB58*-AraOX plants weighed less than those of Col-0 plants under both high- and low-Pi conditions (Figure 3e,f, and Appendix A). The root architecture of *OsMYB58*-AraOX was significantly altered at both Pi levels (Figure 3d), with reductions in the lengths of primary and lateral roots, as well as the number of lateral roots (Figure 3d,g,h, and Appendix A). Pi concentrations in both shoots and roots were significantly lower in *OsMYB58*-AraOX plants compared to Col-0 (Figure 3i,j, and Appendix A). These results indicate that the phenotypic alterations in Arabidopsis plants under both low-Pi and high-Pi conditions in response to *OsMYB58* overexpression were due to disturbed Pi homeostasis.

### 2.3. OsMYB58 Modulates Plant Growth during the Pi Deficiency Response

To characterize the role of OsMYB58 in the Pi starvation response, we generated two types of transgenic rice plants, *OsMYB58* overexpression plants (*OsMYB58*-OX; Figure 4a) via *Agrobacterium*-mediated transformation. We were offered OsMYB58 knock-out mutants (OsMYB58-KO) produced using the T-DNA tagging system from Dr. Jung (Figure 4c) [32]. The expression levels of *OsMYB58* were similar in most *OsMYB58*-OX lines, as revealed through the use of RT-PCR analysis; we randomly selected line #5 for further analysis (Figure 4b). We selected three *OsMYB58*-KO T1 lines based on their growth in a medium containing hygromycin due to the presence of the *hygromycin phosphotransferase* (*HPH*) gene. We performed genotyping PCR (Figure 4d) and RT-PCR (Figure 4e) analysis of these three selected *OsMYB58*-KO lines to confirm the presence of the T-DNA insertion and used line #6 in all our experiments.

To investigate the phenotypic alterations due to *OsMYB58* overexpression or knock-out, we exposed 7-day-old wild-type (WT) *OsMYB58*-OX and *OsMYB58*-KO plants to either high-Pi or low-Pi conditions for 7 days. Shoot and root growth were strongly inhibited in *OsMYB58*-OX plants compared to WT and *OsMYB58*-KO plants under both high-Pi and low-Pi conditions (Figure 4f and Appendix A). The fresh weights of the shoots and roots were significantly reduced in *OsMYB58*-OX plants compared to WT under both conditions (Figure 4g,h, and Appendix A). By contrast, the fresh weights of the shoots or roots were slightly reduced in *OsMYB58*-KO plants compared to WT regardless of Pi concentration (Figure 4g,h, and Appendix A). The shoots and primary roots were significantly shorter in *OsMYB58*-OX than in WT under both high- and low-Pi conditions (Figure 4i,j, and Appendix A). The shoot length was reduced in *OsMYB58*-KO compared to the WT, while the primary root length was similar to the WT under both sets of conditions (Figure 4i,j, and Appendix A). These results indicate that OsMYB58 affects plant growth and developmental processes regardless of the Pi concentrations in plants.

### 2.4. OsMYB58 Represses Root Development in Rice

Pi deficiency significantly suppresses primary root growth by disrupting the root meristem, with effects such as reducing cell elongation and interrupting cell proliferation and cell differentiation [33]. To investigate the changes in root architecture in response to *OsMYB58* overexpression or knock-out, we transferred 3-day-old WT, *OsMYB58*-OX, and *OsMYB58*-KO seedlings to a growth medium containing high or low Pi concentrations and allowed them to grow for 7 days. The root architecture of *OsMYB58*-KO was similar to that of the WT, whereas root growth was dramatically reduced in *OsMYB58*-OX under both high-Pi and low-Pi conditions (Figure 5a). An overall comparison of roots, including seminal roots, lateral roots, and root hairs, indicated that the lengths and numbers of these structures were lower in *OsMYB58*-OX than in the WT, while those in *OsMYB58*-KO were similar to the WT, under both high- and low-Pi conditions (Figure 5b–g and Appendix A). 

To investigate the changes in Pi concentrations in plants in response to *OsMYB58* overexpression or knock-out, we treated 7-day-old WT, *OsMYB58*-OX, and *OsMYB58*-KO seedlings with high Pi and low Pi. The Pi levels in shoots and roots were lower in *OsMYB58*-OX than in the WT under both high-Pi and low-Pi conditions (Figure 6a,b, and Appendix A). However, the Pi concentrations in shoots and roots of *OsMYB58*-KO plants were similar to those of WT plants under both conditions (Figure 6a,b, and Appendix A). These results suggest that OsMYB58 modulates Pi levels by reducing Pi uptake in plants through alteration of root architecture during Pi deficiency responses.

### 2.5. OsMYB58 Is Associated with Pi-Responsive Gene Expression

Pi-responsive genes and Pi transporters (PTs) improve plant development and growth to help maintain Pi homeostasis via Pi-starvation signaling [34]. To investigate whether Pi-responsive and PT genes are regulated by *OsMYB58*, we treated 7-day-old WT, *OsMYB58*-OX, and *OsMYB58*-KO seedlings with high Pi and low Pi and subjected samples from the seedlings to quantitative RT-PCR (qRT-PCR). *OsmiR399a*, *OsmiR399j*, *OsIPS1*, *OsPT2*, and *OsPT4* transcript levels were significantly reduced in the shoots and roots of *OsMYB58*-OX plants compared to the WT (Figure 7). In *OsMYB58*-KO plants, *OsmiR399a*, *OsmiR399j*, and *OsIPS1* were more highly expressed in shoots and roots compared to the WT (Figure 7a–c). The transcript levels of *OsPHO2* and the target genes of *OsmiR399* significantly increased in shoots and roots of *OsMYB58*-OX but slightly decreased in *OsMYB58*-KO compared to the WT (Figure 7d). The expression levels of *OsPT2* and *OsPT4* in shoots and roots of *OsMYB58*-OX plants were similar to those of the WT (Figure 7e,f). Overall, the effects of *OsMYB58* on Pi-responsive and PT gene expression were more varied under low-Pi conditions than under high-Pi conditions (Figure 7). These results suggest that OsMYB58 functions in Pi starvation signaling by regulating the transcription of Pi-responsive genes, especially *OsmiR399*, an important component of PHR-miR399-PHO2-mediated mechanisms.

Finally, to test the transcriptional activity of OsMYB58, we performed a transcriptional activity assay via transient expression using Arabidopsis protoplasts. We generated an effector construct harboring OsMYB58 with the yeast GAL4 DNA binding domain (DBD) and a reporter construct harboring constitutively expressed GUS with four upstream GAL4 DNA binding sites, Gal4 (4X)-D1-D3 (4X); the *LUC* (*luciferase*) gene driven by the constitutive 35S promoter was used as an internal control (Figure 8). We co-transfected Arabidopsis protoplasts with various combinations of effector and reporter constructs and measured GUS activity. The GUS activity was significantly lower in protoplasts harboring the effector and reporter constructs compared to protoplasts harboring empty vector (DBD-vector) alone (Figure 8). These results indicate that OsMYB58 functions as a transcriptional repressor in Pi starvation signaling.

## 3. Discussion

Pi deficiency in soil has adverse effects on plant growth and development, such as damaging root architecture, inhibiting leaf development, and impairing fruit production via Pi starvation signaling [33,35]. Although numerous transcription factors are reported to function in Pi starvation signaling, few functional studies have focused on the roles of rice MYB transcription factors in this process. To date, only five MYB transcription factors have been shown to function in Pi starvation signaling in rice: OsPHR2, OsMYB1, OsMYB2P-1, OsMYB4P, and OsMYB5P [17,18,19,20,36]. Here, we characterized the role of OsMYB58 in Pi starvation signaling, demonstrating that it acts as a transcriptional repressor that regulates the expression of Pi-responsive genes. *OsMYB58* expression strongly increased in response to Pi deficiency. Overexpressing *OsMYB58* inhibited plant growth and root development by impairing Pi homeostasis in shoots and roots of Arabidopsis and rice. *OsmiR399a* and *OsmiR399j* levels strongly decreased in response to *OsMYB58* overexpression in roots under Pi-deficient conditions. Our results demonstrate that OsMYB58 plays an important role in the PHR-miR399-PHO mechanism for root-to-shoot translocation of Pi.

### 3.1. OsMYB58 Is Associated with Plant Response to Pi Deficiency

Plant MYB transcription factors have important roles in regulating many processes in plants, including growth and development, secondary metabolism, phytohormone signaling, pathogen resistance, and abiotic stress tolerance [37]. Many R2R3 MYB transcription factors are associated with transcriptional regulation in response to various environmental stresses, such as salt, drought, cold, and nutrient deficiency [6,7]. Five R2R3 MYB transcription factors have been shown to regulate Pi deficiency responses in rice [17,18,19,20,27,36]. However, the roles of R2R3 MYB transcription factors in Pi starvation signaling to maintain Pi homeostasis in rice are poorly understood. Here, we demonstrated that *OsMYB58* is strongly expressed in shoots and roots under Pi deficiency compared to other nutrient deficiencies (Figure 1). The OsMYB58 protein sequence, which contains an R2R3-MYB domain at the N-terminal region, is similar to that of OsMYB63 (LOC_Os04g50770) in rice and AtMYB58 (AT1G16490)/AtMYB63 (AT1G79180) in Arabidopsis (Figure 2). The expression of OsMYB63 was slightly changed in nitrogen or potassium deficiency responses; however, it was not different in phosphate deficiency responses (Appendix A). Although protein sequences between OsMYB58 and OsMYB63 are very similar, they probably have different functions in nutrient deficiency responses. OsMYB58 was recently shown to regulate the expression of the cellulose synthase gene *OsCesA7* [31]. The correlation network between Pi starvation signaling and lignin biosynthesis by OsMYB58 has not yet been studied, and it is imperative to continue the functional study of OsMYB58 using that and other approaches.

### 3.2. OsMYB58 Inhibits Plant Growth and Development by Disrupting Pi Homeostasis

Overexpressing MYB transcription factor genes in rice reduced the damage caused by Pi deficiency to plant growth and shoot and root development by enhancing Pi uptake [17,18,20,27]. Plants overexpressing *OsPHR2*, *OsMYB2P-1*, *OsMYB4P*, and *OsMYB5P* showed common phenotypes, including increased shoot, primary root, seminal root, and lateral root growth, due to increased Pi levels under Pi-deficient conditions [17,18,20,27]. By contrast, in the current study, *OsMYB58*-overexpressing Arabidopsis and rice plants were hypersensitive, as compared to the WT, to both low and high Pi levels in terms of shoot growth and root architecture (Figure 3, Figure 4 and Figure 5). Overexpressing *OsMYB58* disrupted Pi homeostasis in rice by inhibiting Pi uptake in shoots and roots (Figure 6). These results contrast with those from other studies of Pi-responsive MYB transcription factors. OsSPX1 and OsSPX2, which are not MYB transcription factors, negatively regulate Pi levels, as their overexpression reduced Pi concentrations in shoots [38]. Our results demonstrate that OsMYB58 acts as a negative regulator of plant growth and development based on cellular Pi concentration.

### 3.3. OsMYB58 Is a Negative Regulator of Pi Starvation Signaling

During Pi starvation signaling in rice, the pathway regulating Pi homeostasis is mediated by two major pathways: PHR2-miR399-PHO2 and PHR2-miR827-SPX/MSF [26,39,40]. The PHR2-miR399-PHO2 pathway modulates Pi-regulated plant phenotypes and Pi accumulation, while the PHR2-miR827-SPX/MSF pathway only regulates Pi accumulation without inducing phenotypic alterations in response to Pi deficiency [26]. *OsmiR399*-dependent *OsPHO2* expression influences Pi uptake and remobilization to help maintain Pi homeostasis [41,42]. OsPHO2 acts as a positive regulator of *OsPT2* gene expression in Pi starvation signaling [41,42]. We demonstrated that the expression of *OsmiR399a* and *OsmiR399j* significantly decreased in *OsMYB58*-OX plants, whereas the expression of *OsPHO2* strongly increased under both high-Pi and low-Pi conditions (Figure 7). In addition, OsMYB58 acted as a transcriptional repressor in a transcriptional activity assay (Figure 8). The decline in Pi levels in *OsMYB58*-OX plants was likely caused by the reduced expression of *OsPT2* and *OsPT4* (Figure 6 and Figure 7). These findings demonstrate that OsMYB58 modulates cellular Pi concentrations in shoots and roots through the PHR2-miR399-PHO pathway during Pi starvation signaling. Thus, OsMYB58 functions in OsmiR399-dependent Pi-starvation signaling by negatively regulating Pi homeostasis. Our findings demonstrate that OsMYB58 functions as a negative regulator of OsmiR399-dependent Pi-starvation signaling.

## 4. Materials and Methods

### 4.1. In Silico Analysis

The sequences of *OsMYB58* (*LOC_Os02g46780*) homologs were obtained from NCBI (http://blast.ncbi.nlm.gov/Blast,cgi (accessed on 5 June 2022)). Multiple protein sequence alignment was performed with the Clustal Omega program (https://www.ebi.ac.uk/Tools/msa/clustalo/ (accessed on 11 January 2023)) and MEGA X (https://www.megasoftware.net/ (accessed on 11 January 2023)).

### 4.2. Generation of MYB58 Transgenic Plants

The *OsMYB58:pH7WG2D.1* (*OsMYB58*-OX) constructs were introduced into *Agrobacterium tumefaciens* (EHA105 and GV2260) via electroporation. The transformation of rice and Arabidopsis was performed as previously described [18]. Dr. Ki Hong Jung (Kyung Hee University) provided *OsMYB58* T-DNA tagged knock-out mutant (*OsMYB58*-KO) rice plants [32].

### 4.3. Plant Materials and Growth Conditions

Physiological experiments and the generation of transgenic plants were performed using *Oryza sativa* L. ‘Dongjin’ and *Arabidopsis thaliana* Col-0 plants. Hydroponic experiments were performed in either a high Pi (0.25 mM KH_2_PO_4_) or low Pi (0.0125 mM KH_2_PO_4_) medium as described previously [18]. Rice or Arabidopsis plants were cultivated in growth chambers at 32 °C or 22 °C, respectively, under a 16 h light/8 h dark cycle.

### 4.4. Northern Bolt and Gene Expression Analysis

The total RNA was isolated from 100 mg of rice tissue using TRIzol reagent (Sigma-Aldrich, St. Louis, MO, USA) according to the manufacturer’s instructions for northern blotting and quantitative reverse-transcription PCR (qRT-PCR). The 20 μM sample of total RNA was separated on a 1.2% (*w*/*v*) denaturing formaldehyde agarose gel via gel electrophoresis, transferred to a Hybrid-N^+^ membrane (GE Healthcare, Chicago, IL, USA), and crosslinked using a commercial UV-light crosslinking instrument (UVP). The membrane was hybridized overnight with a [^32^P]-dCTP-labeled probe (Takara, Shiga, Japan) in a solution containing 20% (*w*/*v*) SDS, 20× SSPE, 100 g/L PEG (8000 mwt), 250 mg/L heparin, and 10 mL/L herring sperm DNA (10 mg/mL) at 65 °C. Probes for the *OsMYB58* gene were generated using a primer set designed for its open reading frame. The filters were washed twice in 2× SSC and 0.2% (*w*/*v*) SDS at 65 °C for 10 min, twice in 1× SSC and 0.2% (*w*/*v*) SDS at 65 °C for 10 min, and once in 0.1× SSC and 0.2%(*w*/*v*) SDS at 65 °C for 20 min. The dried membrane was placed on X-ray film at −72 °C for 1 day and then developed and exposed.

For qRT-PCR, cDNA was synthesized using 0.5 μg of total RNA with a 1st Strand cDNA synthesis Kit (Takara, Dalian, China) and subjected to RT-PCR analysis to measure gene expression using qRT-PCR Detection Systems (Bio-Rad, Hercules, CA, USA) following a standard protocol. The primer sequences were designed using NCBI Primer-BLAST and Primer3 (https://bioinfo.ut.ee/primer3/ (accessed on 10 August 2022)) and are provided in Appendix A. PCR (10 µL reactions) was performed using TB Green Premix Ex Taq™ II (Takara, Dalian, China) as follows: 95 °C for 5 min, and 40 cycles of 95 °C for 15 s, 60 °C for 15 s, and 72 °C for 15 s. The relative expression levels in all the samples were automatically calculated and analyzed three times using a CFX384 Real-time PCR Detection System and CFX Manager software ver. 2.0 (Bio-Rad, Hercules, CA, USA) following a standard protocol.

### 4.5. Measurement of Inorganic Pi Content in Plants

After measurement of fresh weight, samples were frozen and dried at 80 °C for 3 days. Inorganic Pi contents were measured as previously described [18].

### 4.6. Transcriptional Activity Assay via Transient Expression of OsMYB58 in Arabidopsis Protoplasts

To investigate the transcriptional activity of OsMYB58, we performed a transcriptional activity assay by introducing constructs harboring reporter genes driven by the *OsMYB58* promoter into Arabidopsis protoplasts prepared from leaf tissues by PEG-mediated transformation and measuring reporter activity, as described previously [18,20].

## 5. Conclusions

In summary, we characterized the role of the transcription factor OsMYB58 in Pi-starvation signaling. OsMYB58 was dramatically upregulated under Pi deficiency compared to other nutrient-deficient conditions. Overexpressing *OsMYB58* in Arabidopsis and rice strongly inhibited plant growth and root development, whereas *OsMYB58*-KO plants showed enhanced growth and development under both high- and low-Pi conditions. In addition, *OsMYB58*-overexpressing transgenic Arabidopsis and rice showed greatly reduced Pi concentrations in both shoots and roots. The transcript levels of *OsmiR399a* and *OsmiR399j* strongly increased in shoots and roots, whereas *OsPHO2* transcript levels decreased in plants overexpressing *OsMYB58*. OsMYB58 is a transcriptional repressor, as determined by a transcriptional activity assay. Our results suggest that *OsMYB58* negatively regulates Pi-starvation signaling in rice in an *OsmiR399*-dependent manner.

## Figures and Tables

**Figure 1 ijms-25-02209-f001:**
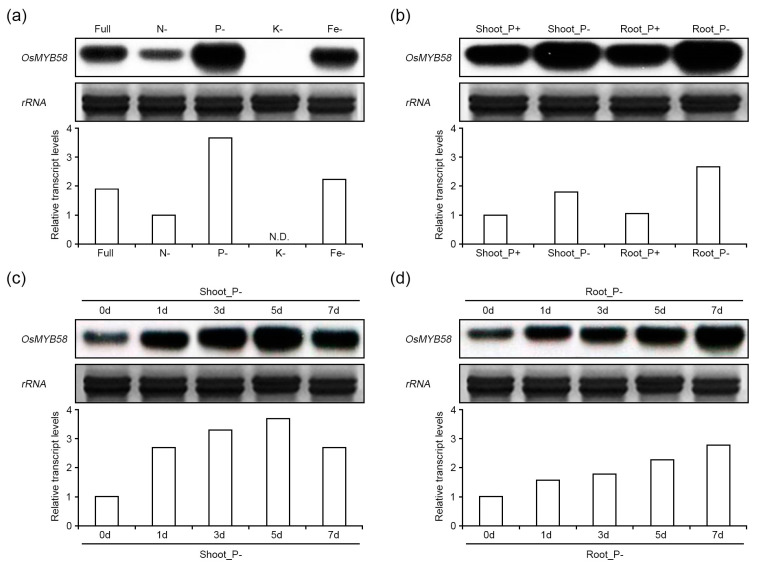
Transcriptional expression of *OsMYB58* in rice under nutrient−deficient conditions. The upper panels of each figure indicated that transcript levels of *OsMYB58* were analyzed using the northern blot. The bottom graphs of each figure indicated that the relative values of band intensity were calculated by *rRNA* intensity. (**a**) Total RNA extracted from rice wild-type plants (*Oryza sativa* L. ‘Dongjin’) growing in various nutrient deficiency conditions. Rice plants were transferred to nitrogen (N−; 0.25 mM), phosphate (P−; 0.0125 mM), potassium (K−; 0.01 mM), or iron (Fe−; 0.01 mM)−deficient media and grown for 3 days. (**b**) Total RNA extracted from shoots and roots of rice plants after 3 days of treatment to high Pi (P+; 0.25 mM KH_2_PO_4_) or low Pi (P−; 0.0125 mM KH_2_PO_4_). (**c**,**d**) Rice samples were treated to low Pi at different time points. The total RNA was extracted from separate parts of the shoot (**c**) and root (**d**) parts of the harvested samples. The *rRNA* was a loading control.

**Figure 2 ijms-25-02209-f002:**
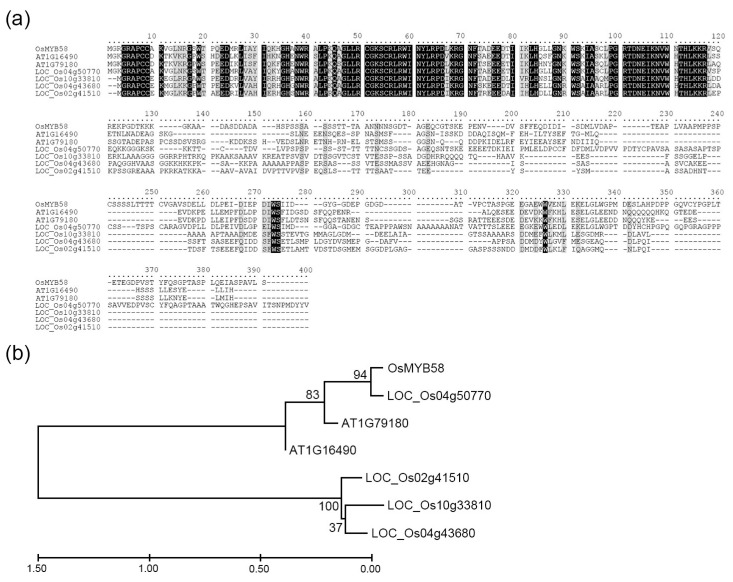
Sequence alignment and phylogenetic tree analysis of R2R3-type MYB transcription factors in Arabidopsis and rice. (**a**) Multiple protein sequence alignment of R2R3-type MYB protein in Arabidopsis and rice was generated by the Clustal Omega program (https://www.ebi.ac.uk/Tools/msa/clustalo/ (accessed on 11 January 2023)). Identical protein sequences are shaded in black, and similar protein sequences are shaded in gray. (**b**) The phylogenetic tree of Arabidopsis and rice MYB proteins was constructed with the Neighbor-Joining method in MEGA X (https://www.megasoftware.net/ (accessed on 11 January 2023)) using R2R3 domain sequences.

**Figure 3 ijms-25-02209-f003:**
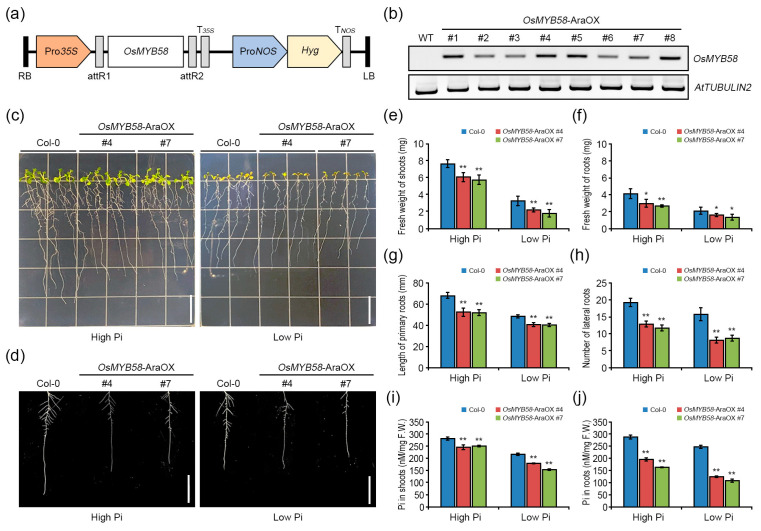
Physiological analysis of *OsMYB58* overexpressing Arabidopsis plants under low-Pi conditions. (**a**) For overexpressing *OsMYB58* in Arabidopsis Col-0 plants, the diagram shows the plasmid DNA construct, including the hygromycin (Hyg) selection marker. (**b**) *OsMYB58* expression in *OsMYB58*-AraOX plants by RT-PCR analysis. Total RNA was extracted from selected *OsMYB58*-AraOX plants using hygromycin. *AtTUBULIN2* is an internal control. (**c**) 4-day-old seedlings of Arabidopsis wild-type (Col-0) and *OsMYB58* overexpressing plants (*OsMYB58*-AraOX) were transferred to medium including high Pi (0.25 mM KH_2_PO_4_) or low Pi (0.0125 mM KH_2_PO_4_) for 7 days and then the photos were taken. The scale bar indicates 1.8 cm. (**d**) Comparison of root architectures between Col-0 and *OsMYB58*-AraOX seedlings depicted in (**c**). The scale bar indicates 1.8 cm. (**e**–**h**) After 7 days to high Pi or low Pi, physiological changes in shoot and root were analyzed using various methodological measurements, including shoot fresh weight (**e**), root fresh weight (**f**), primary root length (**g**), and the number of lateral roots (**h**). (**i**,**j**) Pi concentrations were measured in the shoot (**i**) and root (**j**) of Col-0 and *OsMYB58*-AraOX after treatment to high Pi or low Pi for 7 days. Error bars represent the mean ± standard deviation (SD) of three biological replicates with 10 seedlings for each experiment. Asterisks represent significant differences from the Col-0 (*; 0.01 < *p*-value ≤ 0.05, **; *p*-value ≤ 0.01, Student’s *t*-test).

**Figure 4 ijms-25-02209-f004:**
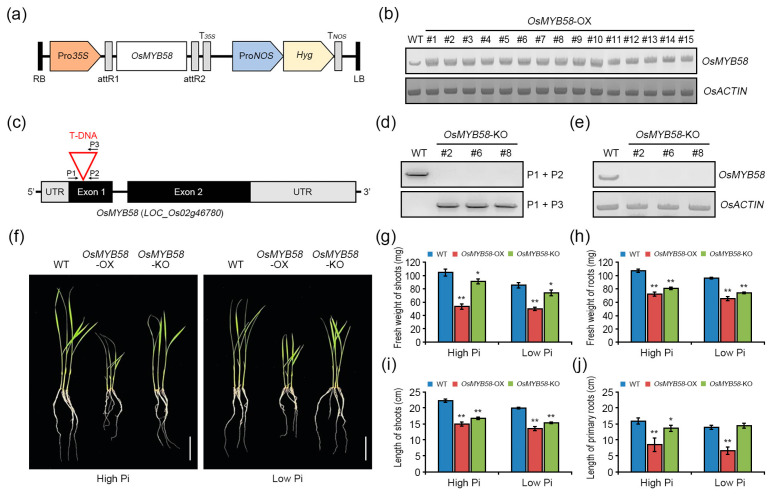
Physiological analysis of *OsMYB58* overexpressing and knock-out mutant rice plants. (**a**) For overexpressing *OsMYB58* in rice plants, the diagram shows the plasmid DNA construct, including the hygromycin (Hyg) selection marker. (**b**) *OsMYB58* expression in *OsMYB58*-OX plants via RT-PCR analysis. Total RNA was extracted from selected *OsMYB58*-OX rice plants using hygromycin. *OsACTIN* is an internal control. (**c**) Schematic diagrams of *OsMYB58* mutation by T-DNA insertion. P1 and P2 mean the specific primers of the *OsMYB58* gene. P3 means the specific primer of exogenous T-DNA. The red color indicates the T-DNA into the *OsMYB58* gene. (**d**) Genotyping PCR analysis in *OsMYB58*-KO plants. For selecting T-DNA-inserted transgenic plants, diagnostic PCR was performed in wild-type (WT) and *OsMYB58*-KO plants using a combination of gene-specific (P1 and P2) or T-DNA-specific (P3) primers. (**e**) *OsMYB58* expression in *OsMYB58*-KO plants by RT-PCR analysis. Total RNA was extracted from selected *OsMYB58*-KO rice plants using hygromycin. *OsACTIN* is an internal control. (**f**) 7-day-old seedlings of rice wild-type (WT), *OsMYB58* overexpressing plants (*OsMYB58*-OX), and *OsMYB58* T-DNA-tagging knock-out mutant (*OsMYB58*-KO) were transferred to medium including high Pi (0.25 mM KH_2_PO_4_) or low Pi (0.0125 mM KH_2_PO_4_) for 7 days and then the photos were taken. The scale bar indicates 5 cm. (**g**–**j**) After 7 days to high Pi or low Pi, physiological changes in the shoot and root were analyzed via various methodological measurements, including fresh weight of shoots (**g**), fresh weight of roots (**h**), length of shoots (**i**), and length of primary roots (**j**). Error bars represent the mean ± standard deviation (SD) of three biological replicates with five seedlings for each experiment. Asterisks represent significant differences from the WT (*; 0.01 < *p*-value ≤ 0.05, **; *p*-value ≤ 0.01, Student’s *t*-test).

**Figure 5 ijms-25-02209-f005:**
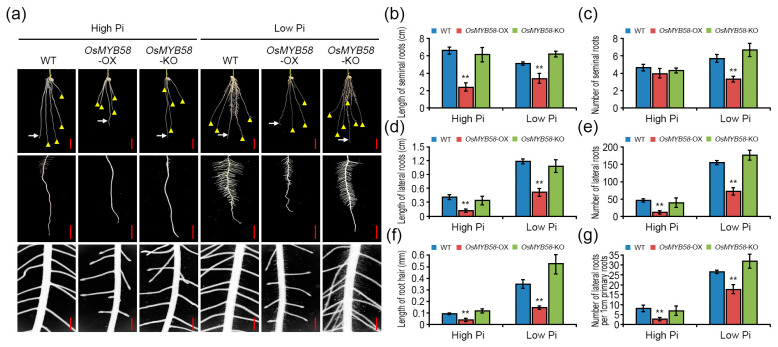
Physiological alteration in root architecture of *OsMYB58*-OX and *OsMYB58*-KO plants. (**a**) 3-day-old seedlings of rice WT, *OsMYB58*-OX, and *OsMYB58*-KO were transferred to medium including high Pi (0.25 mM KH_2_PO_4_) or low Pi (0.0125 mM KH_2_PO_4_) for 7 days and then the photos were taken in root architecture. The white arrow indicated the primary roots, and the yellow arrowhead indicated the seminal roots. Scale bar in upper or middle panels indicated the 1 cm. Scale bar in bottom panels indicated the 0.5 mm. (**b**–**g**) After 7 days to high Pi or low Pi, physiological alteration in root architecture was analyzed using various methodological measurements, including length of seminal roots (**b**), number of seminal roots (**c**), length of lateral roots (**d**), number of lateral roots (**e**), length of root hair (**f**), and number of lateral roots per 1cm primary roots (**g**). Error bars represent the mean ± standard deviation (SD) of three biological replicates with five seedlings for each experiment. Asterisks represent significant differences from the WT (**; *p*-value ≤ 0.01, Student’s *t*-test).

**Figure 6 ijms-25-02209-f006:**
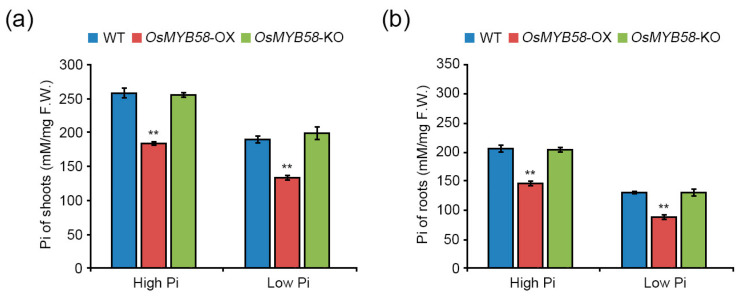
Pi accumulation in *OsMYB58*-OX and *OsMYB58*-KO plants. Pi concentrations were measured in the shoots (**a**) and roots (**b**) of rice WT, *OsMYB58*-OX, and *OsMYB58*-KO plants under high Pi (0.25 mM KH_2_PO_4_) or low Pi (0.0125 mM KH_2_PO_4_) conditions. Error bars represent the mean ± standard deviation (SD) of three biological replicates with five seedlings for each experiment. Asterisks represent significant differences from the WT (**; *p*-value ≤ 0.01, Student’s *t*-test).

**Figure 7 ijms-25-02209-f007:**
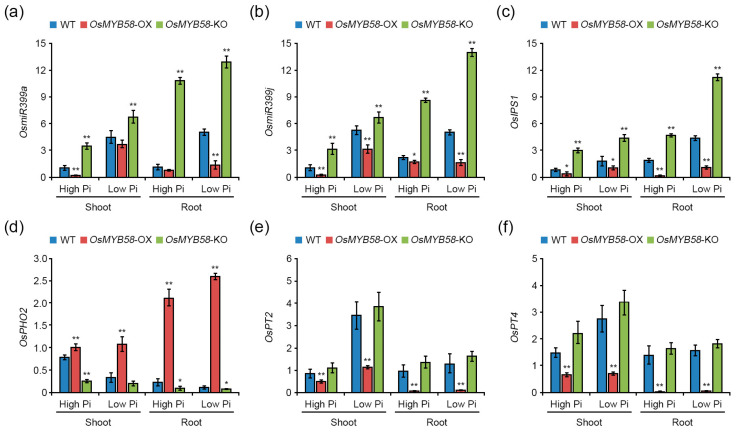
Transcripts comparison of Pi-responsive genes and Pi transporters in *OsMYB58*-OX and *OsMYB58*-KO plants. Seven-day-old seedlings of rice WT, *OsMYB58*-OX, and *OsMYB58*-KO were transferred to medium including high Pi (0.25 mM KH_2_PO_4_) or low Pi (0.0125 mM KH_2_PO_4_) for 7 days. For qRT-PCR analysis, total RNA was extracted from shoots and roots of high Pi- or low Pi-treated plants. qRT-PCR was used to analyze the transcript levels of Pi-responsive genes, such as *OsmiR399a* (**a**), *OsmiR399j* (**b**), *OsIPS1* (**c**), *OsPHO2* (**d**), *OsPT2* (**e**), and *OsPT4* (**f**) using specific primers in Appendix A. Expression of *OsACTIN1* was used for the normalization. Error bars represent the mean ± standard deviation (SD) of three biological replicates. Asterisks represent significant differences from the WT (*; 0.01 < *p*-value ≤ 0.05, **; *p*-value ≤ 0.01, Student’s *t*-test).

**Figure 8 ijms-25-02209-f008:**
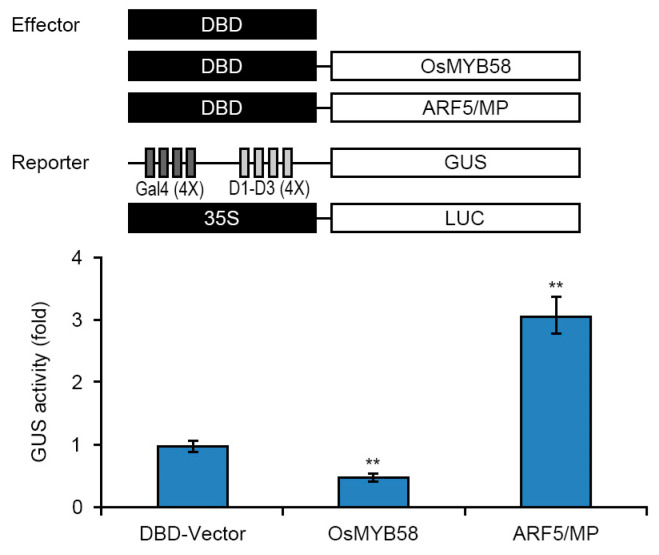
Transcriptional activity of *OsMYB58* in Arabidopsis protoplast transient system. A schematic diagram showed the effector and reporter plasmid DNA used in the transient expression assay. Combinations with each effector, along with two reporters, were co-transfected into protoplasts from 2-week-old Arabidopsis leaves. *ARF5/MP* was used as an experimental positive control, and *35S:LUC* plasmid DNA was used as an internal control. After normalization via LUC activity, GUS activity in each transfected protoplast sample was calculated. Error bars represent the mean ± standard deviation (SD) of three biological replicates. Asterisks represent significant differences from the BD-vector (**; *p*-value ≤ 0.01, Student’s *t*-test).

## Data Availability

Data contained within the article.

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
