# Peer review of "OsMYB58 Negatively Regulates Plant Growth and Development by Regulating Phosphate Homeostasis"

_ijms, 2024, doi:10.3390/ijms25042209_

Round 1
Reviewer 1 Report
Comments and Suggestions for Authors
The work can be accepted but following a change in the introduction because it is very confusing. For me it must be simplified and be simpler and clearer in defining the objectives of the work and the reason for this study. I would like to read what this type of research can be used for from an application point of view.
Author Response
[Reviewer 1]
The work can be accepted but following a change in the introduction because it is very confusing. For me it must be simplified and be simpler and clearer in defining the objectives of the work and the reason for this study. I would like to read what this type of research can be used for from an application point of view.
[Response]
Thank you for your comment. We arranged the introduction and described some of the details in the introduction part of the revised manuscripts.
Reviewer 2 Report
Comments and Suggestions for Authors
The manuscript by Baek et al. suggests an involvement of OsMYB58 in phosphate homeostasis in rice. Their study appears interesting, but I have several concerns on their manuscript as follows.
1. In Figure 1, reproducibility of their data is unknown. Did they perform these assays only once? Multiple trials and confirmation of reproducibility is essential in quantitative levels.
2. They stated that they generated T-DNA insertion knockout mutants in this study. However, the method for obtaining the mutants is missing in the manuscript. It is quite difficult or almost impossible to generate mutants with T-DNA insertions in a gene of interest by targeting the gene. Generally, T-DNA insertion mutants are provided from or identified in large-scale T-DNA insertion mutant libraries. How have they obtained T-DNA insertion mutants for OsMYB58?
3. The most important criticism in this study is the lack of reproducibility of overexpression lines and mutant lines. The manuscript only showed one line for each OsMYB58-OX and OsMYB58-KO construct. Thus, their analysis cannot exclude possibilities of artifacts caused by T-DNA insertions in genomic regions which are unrelated to the function of OsMYB58. Investigations in multiple lines are needed for overexpression and mutation analyses.
4. The manuscript concluded that OsMYB58 modulates Pi levels by reducing Pi uptake. However, no direct evidence is demonstrated for reducing Pi uptake, because Pi content would be influenced by several factors including internal Pi remobilization in addition to Pi uptake. How will the transgenic plants behave on media that completely lacks phosphate?
5. OsMYB63 is a close homolog of OsMYB58. Is the transcription of OsMYB63 affected by Pi deficiency?
6. The molecular function of the PHO2 protein should be described in the introduction section.
7. What is "MBS motifs" in line 60?
Author Response
[Reviewer 2]
- In Figure 1, reproducibility of their data is unknown. Did they perform these assays only once? Multiple trials and confirmation of reproducibility is essential in quantitative levels.
[Response]
Thank you for your comments. In Figure 1, OsMYB58 was more highly expressed in phosphate deficiency conditions than in other nutrient deficiency conditions. To confirm the transcript levels of OsMYB58 in nutrient deficiency conditions, we performed the qRT-PCR analysis using another rice cultivar ‘Ilmi’ plants (Supplementary Figure S1). Like Figure 1 results, the transcript level of OsMYB58 was highest in phosphate deficiency conditions compared to other conditions. We added the results of qRT-PCR analysis in Supplementary Figure S1 of the revised manuscripts.
- They stated that they generated T-DNA insertion knockout mutants in this study. However, the method for obtaining the mutants is missing in the manuscript. It is quite difficult or almost impossible to generate mutants with T-DNA insertions in a gene of interest by targeting the gene. Generally, T-DNA insertion mutants are provided from or identified in large-scale T-DNA insertion mutant libraries. How have they obtained T-DNA insertion mutants for OsMYB58?
[Response]
Thank you for your comments. We were provided with OsMYB58-KO rice plants by Professor Ki Hong Jung’s lab. Jeon et al., 2000 reported the technical methods for screening from large-scale T-DNA insertion mutant libraries. To confirm the mutation of the OsMYB58 gene in OsMYB58-KO, we performed the genotyping PCR (diagnostic PCR) and RT-PCR analysis in Figure 4a-d. We added the “Jeon et al., 2000” reference and modified the Figure S4 of the revised manuscripts.
References
Jeon, J.S.; Lee, S.; Jung, K.H.; Jun, S.H.; Jeong, D.H.; Lee, J.; Kim, C.; Jang, S.; Yang, K.; Nam, J.; An, K.; Han, M.J.; Sung, R.J.; Choi, H.S.; Yu, J.H.; Choi, J.H.; Cho, S.Y.; Cha, S.S.; Kim, S.I.; An, G. DNA insertional mutagenesis for functional genomics in rice. Plant J. 2000, 22, 561-70. doi: 10.1046/j.1365-313x.2000.00767.x.
- The most important criticism in this study is the lack of reproducibility of overexpression lines and mutant lines. The manuscript only showed one line for each OsMYB58-OX and OsMYB58-KO construct. Thus, their analysis cannot exclude possibilities of artifacts caused by T-DNA insertions in genomic regions which are unrelated to the function of OsMYB58. Investigations in multiple lines are needed for overexpression and mutation analyses.
[Response]
Thank you for your comments. We analyzed the multiple lines of OsMYB58-AraOX, OsMYB58-OX, and OsMYB58-KO before the submission of our manuscripts. We showed the representative line of OsMYB58-AraOX, OsMYB58-OX, and OsMYB58-KO in the manuscripts. In the revised manuscripts, we added the organized tables about the results of the biomass analysis in the OsMYB58-AraOX, OsMYB58-OX, and OsMYB58-KO plants under high Pi and low Pi conditions (Supplementary Table S2 and S3).
- The manuscript concluded that OsMYB58 modulates Pi levels by reducing Pi uptake. However, no direct evidence is demonstrated for reducing Pi uptake, because Pi content would be influenced by several factors including internal Pi remobilization in addition to Pi uptake. How will the transgenic plants behave on media that completely lacks phosphate?
[Response]
Thank you very much for your comments. Although we did not perform the physiological analysis of OsMYB58 plants in Pi deficiency conditions, we can predict similar phenotypes of OsMYB58 plants in the low Pi and Pi deficiency conditions when we referred to the previously reported Baek et al., 2013. We described this information in the discussion parts of the revised manuscripts.
References
Baek, D.; Kim, M.C.; Chun, H.J.; Kang, S.; Park, H.C.; Shin, G.; Park, J.; Shen, M.; Hong, H.; Kim, W.Y.; Kim, D.H.; Lee, S.Y.; Bressan, R.A.; Bohnert, H.J.; Yun, D.J. Regulation of miR399f transcription by AtMYB2 affects phosphate starvation responses in Arabidopsis. Plant Physiol. 2013, 161, 362-373. doi: 10.1104/pp.112.205922.
- OsMYB63 is a close homolog of OsMYB58. Is the transcription of OsMYB63 affected by Pi deficiency?
[Response]
Thank you for your comments. To investigate the transcript levels of OsMYB63 in nutrient deficiency conditions, we performed the qRT-PCR analysis in rice cultivar ‘Ilmi’ plants (Supplementary Figure S2). The OsMYB63 was slightly expressed in nitrogen and potassium deficiency conditions, however, it was different in phosphate deficiency conditions. We added the results of qRT-PCR analysis in Supplementary Figure S2 and described in the discussion parts of the revised manuscripts.
- The molecular function of the PHO2 protein should be described in the introduction section.
[Response]
Thank you for your comments. We added the molecular function of PHO2 protein in the introduction parts of the revised manuscripts.
- What is "MBS motifs" in line 60?
[Response]
Thank you. We added the full name of “MBS motif” in the revised manuscripts.
Reviewer 3 Report
Comments and Suggestions for Authors
Comments for the manuscript entitled "OsMYB58 negatively regulates phosphate acquisition via OsmiR399 - dependent phosphate starvation signaling in rice" submitted by Dongwon Baek et al.
This paper focuses on the role of the transcription factor OsMYB in rice in phosphate (Pi) starvation signalling. So far there are only a few studies on the role of MYB transcription factors in rice, showing that there are 5 R2R3MYB transcription factors that regulate the Pi deficiency response in rice. But the roles of these R2R3MYB in signaling Pi starvation in rice have not been fully understood.
The overall merit of the present study is that it demonstrated that OsMYB58 is strongly expressed in shoots and roots under Pi deficiency compared to other nutrient deficiencies. Physiological and biological analyses under high Pi and low Pi conditions have shown that OsMYB58 plays a key role in maintaining Pi homeostasis.
To explore the roles of OsMYB58 in the Pi starvation response, research was conducted on wild rice (Oryza sativa L. 'Dongjin'), two Arabidopsis lines (Col-0 and IsMYB58-AraOX). To show that OsMYB58 modulates plant growth during Pi deficiency response, investigations were done on two types of transgenic rice plants (OsMYB58-OX) and T-DNA-tagged knock-out mutants (OsMYB58-KO). The results suggest that OsMYB58 modulates Pi levels by reducing Pi uptake in plants by altering root architecture during Pi deficiency responses. Thus, OsMYB58 has been shown to act as a transcriptional repressor that regulates the expression of environmental phosphorous-responsive genes.
In Pi deficiency, OsMYB58 expression increased strongly, producing a cascade of effects: impaired Pi homeostasis in shoots and roots, inhibition of plant growth and root development, decreased expression levels of OsmiR399a and OsmiR399j (microRNAs in rice).
In conclusion, the results demonstrate that OsMYB58 acts as a negative regulator of plant growth and development based on cellular Pi concentration.
My comments are below:
1. No dot after title!
2. This title of this manuscript should be improved, more accessible to readers.
3. In line 57, it is not necessary to write Arabidopsis on parentheses.
4. Figures S1, S2, S3, S4 and Table S1 should be placed in the text of the paper. Figure S1 (a), should be clearer. Sequences are written in very small letters.
5. Provide a concrete and valid explanation for your statement: "Understanding the role of OsMYB58 in PHR-miR399-PHO2-dependent mechanisms during Pi-starvation signaling will help enhance crop yields and growth in nutrient-poor soils".
I wish you good luck getting your manuscript published!
Author Response
[Reviewer 3]
- No dot after title!
[Response]
Thank you for your comment. We removed the dot after the title in the revised manuscript.
- This title of this manuscript should be improved, more accessible to readers.
[Response]
Thank you for your comment. We changed the title in the revised manuscript.
- In line 57, it is not necessary to write Arabidopsis on parentheses.
[Response]
Thank you for your comment. We removed the “Arabidopsis on parentheses” in the revised manuscript.
- Figures S1, S2, S3, S4 and Table S1 should be placed in the text of the paper. Figure S1 (a), should be clearer. Sequences are written in very small letters.
[Response]
Thank you for your comment. We edited the main figures, including supplementary figures, in the revised manuscript. Also, we modified the larger size of a letter in protein sequences in the revised Figure 1.
- Provide a concrete and valid explanation for your statement: "Understanding the role of OsMYB58 in PHR-miR399-PHO2-dependent mechanisms during Pi-starvation signaling will help enhance crop yields and growth in nutrient-poor soils".
[Response]
Thank you for your comment. We described in more detail about your comments in the revised manuscripts.
I wish you good luck getting your manuscript published!
[Response]
Thank you very much for your comment.
Round 2
Reviewer 2 Report
Comments and Suggestions for Authors
I have confirmed that the authors answered to all of my concerns. Only an additional comment to the revised manuscript is that the sentences in lines 203-205 may be
"To characterize the role of OsMYB58 in the Pi-starvation response, we generated OsMYB58 overexpression plants (OsMYB58-OX; Figure 4a) via Agrobacterium-mediated transformation. In addition, we were offered OsMYB58 knock-out mutants (OsMYB58-KO) by the T-DNA tagging system from Dr. Jung (Figure 4c) [32]".